# Short- and Long-Term Impacts of the COVID-19 Pandemic on Suicide-Related Mental Health in Korean Adolescents

**DOI:** 10.3390/ijerph191811491

**Published:** 2022-09-13

**Authors:** Byungha Lee, Jung Su Hong

**Affiliations:** 1Shattuck-St. Mary’s School, Faribault, MN 55021, USA; 2Kunsan College of Nursing, 7, Donggaejung-gil, Kunsan-si 54068, Jeollabuk-do, Korea

**Keywords:** COVID-19, adolescents, depression, suicide, Korea

## Abstract

This study investigated the short-term (in 2020) and long-term (in 2021) impacts of the COVID-19 pandemic on suicide-related characteristics in Korean adolescents in comparison with the pre-pandemic period (in 2019) and examined the factors associated with those impacts. Secondary data of the cross-sectional 15th–17th (2019–2021) Korea Youth Risk Behavior Web-based Survey targeting adolescents in school were utilized. The proportions of adolescents with depression, suicidal ideation, suicide planning, and suicide attempts were 26.5%, 12.2%, 3.7%, and 2.4%, respectively. Following an adjustment, depression, suicidal ideation, and suicide attempts significantly improved in the short term and depression and suicide attempts improved significantly in the long term (i.e., 2021), albeit to a lesser degree. The associated variables in the short-term analysis (i.e., 2020) reflected the socioeconomic vulnerabilities (e.g., lower household socioeconomic status [SES], unhealthy status, and unhealthy behaviors), as well as the socioeconomically favorable indicators (e.g., high household SES and high academic achievement). In the long-term, suicidal ideation was no longer associated with a lack of engagement in hand washing, and suicide attempts were no longer associated with the amount of internet time used for studying. For a successful suicide prevention, it is necessary to develop in-school interventions that address the relevant factors identified in this study and the community-based interventions that target out-of-school adolescents.

## 1. Introduction

Suicide has an immense burden on global public health. Suicide is a particularly serious problem among adolescents, as it was the third leading cause of death among 10- to 19-year-olds globally as of 2015 [1]. In addition, the suicide rate of South Korean adolescents was the fourth-highest among the Organization for Economic Co-operation and Development countries, and it consistently increased through 2019 [2]. Following the outbreak of the coronavirus disease 2019 (COVID-19), which was first reported at the end of 2019, quarantine policies, such as school closures, as part of the social distancing policies, commenced; in 2020 and 2021, these policies socially isolated adolescents and led them to worry about illness and the economic impacts on their families [3,4]. This has caused health workers to be concerned about mental health among adolescents, as a vulnerable population.

Depression is one of the strongest predictors of suicidal ideation, but not necessarily of suicide attempts [5]. Furthermore, the development of suicidal ideation and the progression from ideation to suicide attempts are distinct phenomena, as demonstrated by the ideation-to-action framework [5]. Therefore, these psychological variables both need to be examined in order to prevent suicide. However, according to review studies on depression among adolescents during the COVID-19 pandemic, most studies showed that depression among adolescents worsened in 2020 after the outbreak of COVID-19 [4,6]. Although contradictory results were found, as a few studies reported improvements [7,8]. Furthermore, a review [9] on suicidal ideation and suicide attempts demonstrated that among all six studies on suicidal ideation, only three reported increased estimates of suicidal ideation during the COVID-19 pandemic in Canada, China, and the United States. Suicide planning and attempts were hardly found [10], and inconsistent suicide rates in 2020 have been reported in different countries [2].

The diversity of these study results could be attributed to the differences among the studies in the measurement time and countries, as well as the aspects of the study methods, such as study sampling and measurement tools [11,12]. In a meta-analysis of longitudinal studies of predominantly European and North American adults, the worst mental health was found in the early stages of the pandemic (March to April 2020), whereas improvements were found later in the same year. The psychological adaptation and resilience in the later stage would be attributed to the fact that those findings were reported during a later stage of the pandemic [12]. Furthermore, countries have distinctive characteristics in terms of socioeconomic factors, health status, healthy lifestyles, and health care services, all of which are associated with mental health [11].

Therefore, in order to derive reliable results regarding the suicide-related characteristics of adolescents during the COVID-19 pandemic, this study analyzed the representative data from Korean adolescents before the pandemic (in 2019), in the short-term period after the start of the pandemic (2020), and in the long-term period of the pandemic (in 2021). The variables that influenced suicidality among adolescents were also considered. The results of this study will contribute basic data to produce policies helping to prevent adolescent suicide.

## 2. Materials and Methods

### 2.1. Study Design

This study is a secondary analysis using raw data from the 15th (2019), 16th (2020), and 17th (2021) Youth Health Behavior Online Survey (KYRBS) to identify differences in the suicide-related characteristics of adolescents in Korea before (2019) and during the COVID-19 pandemic, in the short term (2020) and long term (2021).

### 2.2. Participants

The KYRBS data provided by the Korea Centers for Disease Control and Prevention Agency (KDCA) were analyzed in order to obtain insights into the status of the health behavior of young people in the Republic of Korea, to calculate the health indicators necessary for the planning and evaluation of youth health promotion projects, and to facilitate quantitative comparisons of youth health indicators between countries. The KYRBS was officially approved by the Statistics Korea, and the Institutional Review Board of the KDCA has waived the need for the KYRBS’s ethics approval under the Bioethics & Safety Act, and it has been made available to the public online. All study participants completed the survey using a computer under the condition of anonymity. The KYRBS is a cross-sectional study conducted annually with students aged 12–18 years at 400 middle schools and 400 high schools, using a multi-stage cluster sampling design to acquire the representative data [13]. As the sampling frame for the sample design, data from middle and high schools nationwide as of April 2019, April 2020, and April 2021 were used. The sampling consisted of three steps: population stratification, sample allocation, and sampling. In the population stratification stage, in order to minimize the sampling error, the regional group and school level were used as stratification variables. In the sample allocation step, the proportional distribution method was applied to ensure that the population composition ratio for each stratified variable and the sample composition ratio were consistent. Stratified cluster sampling was used, and the primary sampling unit was a school and the secondary sampling unit was a class. The 2019 KYRBS was conducted from June to July 2019, and 57,303 students participated in the study. The 2020 KYRBS was conducted from August to November 2020, and 54,948 students participated. The 2021 KYRBS was conducted from August to November 2021, and 54,848 students participated. Therefore, among the 167,099 respondents from the three years (2019–2021), the data from 159,189 students were analyzed, with the exclusion of 7910 respondents with missing responses on the suicide-related variables. More detailed material on the data, data collection, and the survey procedure is available online (https://knhanes.kdca.go.kr/knhanes/main.do) (accessed on 3 June 2022).

### 2.3. Outcomes and Covariates

As suicide-related variables, depression, suicidal ideation, suicide planning, and suicide attempt were analyzed. Depression was assessed based on the participants’ experience of depressive mood through the question “During the last 12 months, have you ever had enough sad or desperate times to stop your daily activities for two weeks?” The experience of a depressive mood persisting for more than two weeks is one of the core symptoms of a major depressive disorder, according to the Diagnostic and Statistical Manual of Mental Disorders, fifth Edition (DSM-V), and it is also used as a screening question [14,15]. Suicidal ideation, suicide planning, and suicide attempts were assessed by questions asking whether suicidal thoughts, suicide planning, and suicide attempts had occurred in the past 12 months (with answers of yes or no).

As independent variables, general and health-related characteristics were included. Data on the following general characteristics were analyzed: age, school type (middle school, high school), sex, academic achievement, type of residence, and household socioeconomic status (SES). Academic achievement and household socioeconomic status (SES) were assessed by self-reported responses and reclassified as high (high or high middle), middle, and low (middle low or low). The type of residence was categorized according to whether the participants lived with family (yes or no).

Several health-related characteristics were analyzed, as follows: subjective health status, perceived body image, and health behaviors including breakfast skipping, physical activity, internet usage time for study and leisure, obesity, personal hygiene, current smoking, and current drinking. The subjective health status and perceived body image were subjectively evaluated by the respondents, whose responses were reclassified into healthy (very healthy or healthy), average, and unhealthy (unhealthy or very unhealthy), and thin (very thin or thin), average, and fat (fat or very fat). The participants reported the frequency of their breakfast consumption and the number of days of physical activity in the preceding seven days. Breakfast skipping was reclassified as having eaten breakfast on five days or less and having eaten breakfast on more than five days, except for those who consumed only milk or juice for breakfast in the last seven days. The responses on physical activity were recategorized as engaging in physical activity on three or more days or on less than three days in the last seven days, in which their heart rate was higher than usual or whether they engaged in 60 min or more of physical activity that left them breathless, regardless of the type of physical activity. Internet usage time was assessed in terms of how many hours per day on average the participants used the internet for study or leisure in the last seven days. According to the Korean Obesity Society’s Obesity Treatment Guidelines [16], based on the 2017 standard growth chart for children and adolescents, the body mass index was classified as indicating underweight, normal, overweight, and obese if it was less than in the 5th percentile, from the 5th percentile to less than in the 85th percentile, from the 85th percentile to less than in the 95th percentile, and in the 95th or above percentile, respectively. Personal hygiene was measured by how often hands were washed with soap in the past seven days, with a total possible score of 20 points (before meals in school, after using the bathroom in school, before meals at home, after using the bathroom at home, and after coming home, with responses of “always”, “mostly”, “sometimes”, and “never” that corresponded to scores of 1 through 4), and responses were reclassified as a total score of 10 or higher or less than 10. Current smoking was assessed by whether participants had smoked more than one cigarette a day in the last 30 days (yes or no), and current drinking was assessed by whether participants had drunk more than one drink in the last 30 days (yes or no).

### 2.4. Statistical Analysis

The KYRBS used a stratified multi-stage cluster-sampling design to provide the representative estimates of the Korean population. The complex sample frequency analysis was performed to calculate the unweighted frequencies and weighted percentages, and the estimated mean and standard error were calculated using a complex-sample descriptive statistical analysis. The general and health-related characteristics were analyzed through a combined sample frequency analysis by year and a complex-sample descriptive statistical analysis. The yearly differences in depression, suicidal ideation, suicide planning and suicide attempts were tested using the Rao–Scott test. The differences in depression, suicidal ideation, suicide attempts, and suicide plans according to general and health-related characteristics, were analyzed with the Rao–Scott test and the combined-sample t-test. The impacts of the year and the general and health-related characteristics on depression, suicidal ideation, suicide planning, and suicide attempts were analyzed in a complex sample using the logistic regression in comparison with 2019. The statistical analysis used SPSS version 23.0, with a significance level of 5%.

## 3. Results

### 3.1. General and Health-Related Characteristics

The general and health-related characteristics of the study population according to the survey years from 2019–2021 are presented in Table 1. This study population was about 15 years old on average and contained more boys than girls. The following characteristics predominated: a low level of academic achievement, living with family, residing in a middle-SES household, being healthy, and perceiving their bodies as fat. Regarding the health behaviors, 20.6% skipped breakfast, 14.4% engaged in physical activity, the average internet use time was 448.74 min for study and 201.53 min for leisure, 9.8% were overweight and 12.3% were obese, 37.3% had a score of at least 10 for personal hygiene, 5.5% smoked, and 11.8% drank alcohol.

### 3.2. Suicide-Related Characteristics and Their Differences from 2019 to 2021

The suicide-related characteristics and their differences from 2019 to 2021 are presented in Table 2. Depression was present among 26.5% of this study population, and 12.2%, 3.7%, and 2.4% reported suicidal ideation, suicide planning, and suicide attempts, respectively. Compared with 2019, in 2020 depression, suicidal ideation, and suicide attempts showed significant improvements, but suicide planning showed no significant differences, and in 2021, depression and suicide attempts showed significant differences by improving, but suicidal ideation no longer showed any significant differences.

### 3.3. Differences from 2019 to 2021 in the Suicide-Related Variables and the Associated Factors

The statistical differences in the general and health-related characteristics according to the suicide-related variables are represented in Table 3. The 2020 and 2021 suicide-related variables, the general and health-related variables, and the relationships among those variables, in comparison with the data from 2019, are presented in Table 4 and Table 5, respectively. Following an adjustment for the general and health-related characteristics, compared with 2019, significant improvements were found in depression, suicidal ideation, and suicide attempts (odds ratio [OR] = 0.88, 95% confidence interval [CI] = 0.85–0.91, *p* < 0.001, OR = 0.86, 95% CI = 0.81–0.90, *p* < 0.001, and OR = 0.74, 95% CI = 0.67–0.82, *p* < 0.001, respectively). In 2021, these improvements were sustained for depression and suicide attempts, albeit to a lesser degree (OR = 0.91, 95% CI = 0.88–0.95, *p* < 0.001, and OR = 0.80, 95% CI = 0.72–0.87, *p* < 0.001, respectively).

Depression was significantly associated in 2020 with older age, being in middle school, the female gender, having a middle or high academic achievement compared with a low achievement, not living with family, a low or high household SES, an average or unhealthy subjective health status compared with a healthy status, a thin or fat perceived body image compared with an average body image, skipping breakfast, engaging in physical activity, internet use time for study and leisure, being at a normal weight compared with being underweight, smoking, and alcohol drinking. The similar associations found in 2021 are all similar to that of 2020, but significantly lower odds for depression were also found among those who were obese and who did not regularly engage in personal hygiene.

Suicidal ideation showed significant associations, in 2020, with younger age, being in middle school, the female gender, having a high academic achievement compared with a low achievement, not living with family, a high household SES compared with a low SES, an average or unhealthy subjective health status compared with a healthy status, a thin or fat perceived body image compared with an average body image, skipping breakfast, engaging in physical activity, internet use time for study and leisure, engaging in personal hygiene, smoking, and alcohol drinking. The significant associations observed in 2020 were maintained in 2021, except for personal hygiene, which no longer showed a statistically significant association in the 2021 data.

Suicidal planning was significantly associated in 2020 with younger age, being in middle school, female, having a high academic achievement compared with a low achievement, not living with family, a low or high household SES compared with an average SES, an average and unhealthy subjective health status compared with healthy status, a fat perceived body image compared with an average body image, skipping breakfast, engaging in physical activity, internet use time for study and leisure, smoking, and alcohol drinking. These associations all remained consistent from 2020 to 2021.

Suicide attempts showed significant associations in 2020 with younger age, being in middle school, female, having a middle or high academic achievement compared with a low achievement, not living with family, a low or high household SES compared with an average SES, an average or unhealthy subjective health status compared with a healthy status, a fat perceived body image compared with an average body image, skipping breakfast, engaging in physical activity, internet use time for leisure, smoking, and alcohol drinking. These associations persisted in 2021, with the exception of internet use time.

## 4. Discussion

This study investigated the mental health status in terms of suicide-related variables and their associations with a range of variables among a representative sample of Korean adolescents during the COVID-19 pandemic. Since the government implemented quarantine policies from 2020 to 2021, this study examined the short-term (6 months) and long-term (18 months) impacts of those policies. The suicide-related variables were found to improve to a greater degree in the short-term, while in the long-term, those improvements were attenuated or disappeared. Both short-term and long-term associations were found for several general characteristics (younger age, female gender, higher academic achievement, not living with family, low or high household SES) and health-related characteristics (unhealthy status, unfavorable perceived self-image, skipping breakfast, engaging in physical activity, internet use time for study and leisure, smoking, and alcohol drinking). Internet usage time for study and personal hygiene showed short-term, but no long-term associations.

In order to better understand suicidality among adolescents, this study examined depression, suicidal ideation, suicide planning, and suicide attempts. Although direct comparisons are difficult due to the diversity among different countries, study samples, and measurement tools [11,12], it is worth noting that the proportions of depression, suicidal ideation, suicide planning, and suicide attempts were 26.5%, 12.2%, 3.7%, and 2.4%, respectively, among this study population. The proportion of participants with depression significantly improved, to a greater extent in the short term than in the long term, compared to before the pandemic, and similar trends were found for other suicide-related variables. We attribute these phenomena to academic stress mitigation, reductions in interpersonal relationship stress, and being in a convenient and comfortable home environment. Due to the lack of preparation for online learning in school after the sudden school closures, the decrease in the number of school activities and assessments would relieve academic stress among Korean students [17]. As schools closed, the attenuated conflicts with peers and bullying in school would relieve interpersonal relationship stress [18,19]. It is also thought that the comfort of home and the financial support for study—that is, the provision of devices for distance learning by schools and local governments—and the economic support for socially and economically vulnerable groups, are likely to have been beneficial for mental health [20,21,22].

However, it should be noted that suicidal ideation worsened in the long term (i.e., 2021), again reaching levels similar to those before the onset of COVID-19. In addition, although suicide attempts improved compared to pre-COVID-19 in this study, the lack of consistent long-term improvements in suicidal ideation and suicide planning, combined with the fact that the actual acts of suicide among adolescents could not be examined in this study, there is no indication that the suicide rate has decreased [2]. Instead, according to a report presented by the Ministry of Health and Welfare in July 2021 [23], while deaths at other ages decreased, among men under the age of 30, the death rate increased through 2020, and this increase was substantially attributable to suicide due to mental health problems. In particular, among teenage males, the increase in the proportion of deaths due to suicide was the highest in 2020 compared with 2019, at 18.8%. The difference between those findings regarding the suicide rate and the results of this study can be explained by the fact that the subjects of this study were adolescents attending school, whereas the number of domestic out-of-school adolescents who did not adapt to school and therefore were vulnerable to mental health issues was 231,823 (4.1%), as of 2020 [24]. This discrepancy seems to be due to the difficulty in managing suicide in out-of-school adolescents. Therefore, a community-based suicide prevention program targeting youth is urgently needed.

Higher rates of the suicide-related variables were found among the more socioeconomically vulnerable population, as exemplified by the observed associations with younger age, being female, not residing with family, having a lower household SES, being unhealthy, and engaging in unhealthy behaviors such as having an irregular diet, inappropriate internet use, smoking, and consuming alcohol, as has been documented elsewhere [3,18,25]. However, unlike other countries, the present study found that adolescents with a high household SES and a high academic achievement would be more vulnerable to poor mental health by limited learning through online learning, as a result of their high academic enthusiasm [21]. These variables showed stronger associations with suicidal ideation during the long-term period of the COVID-19 pandemic. Furthermore, the perceived body image as fat was more strongly associated with mental health than with the actual body status (e.g., obesity) in this study, which seems to be because adolescents pay more attention to their appearance than to health itself [26]. Contrary to the results of other studies [25,27], adolescents who exercised were more vulnerable to poor mental health, in this study. This seems to have been due to the restrictions imposed through social distancing quarantine policies, which prevented active youth from performing as many activities as they wanted and requiring them to exercise alone and/or with a mask; however, this possibility warrants further study in the future [27]. In the long-term period of COVID-19, adolescents with suicidal ideation no longer complied as frequently with the representative quarantine guidelines such as hand washing, which were implemented in the short term. Moreover, adolescents with suicide planning and attempts were more likely to use the internet for leisure for large amounts of time rather than for study [28]. It seems that they lost interest in studying. Community health managers, school officials, and parents should work together to develop policies for interventions by understanding these suicide-related characteristics of Korean adolescents. Furthermore, future studies on the identification of the mental health status including out-of-school adolescents and the related factors are required.

This study has a few limitations. The outcome variables and covariate variables were all self-rated and constructed by a single corresponding survey question, and thus subject to measurement error [29]. Therefore, caution is needed to avoid overinterpreting the results of this study. More standardized survey tools, especially for outcome variables, need to be considered in future research. However, this study also has several strengths. First, this study presents the first comparison of long-term COVID-19 impacts on the suicide-related variables and their associations among Korean adolescents. Second, this study analyzed a nationally representative sample of Korean adolescents.

## 5. Conclusions

Korean adolescents’ depression, suicidal ideation, and suicide attempts improved in the short term after COVID-19 outbreak, but these variables deteriorated somewhat or even reverted to pre-pandemic levels in the longer term, suggesting that there may be concerns about increasing the suicide rates among adolescents in the future. The associated variables were associated with socioeconomic vulnerabilities (e.g., such as younger age, female gender, not residing with family, lower household SES, unhealthy subject status, and unhealthy behaviors such as an irregular diet, inappropriate internet use, smoking, and alcohol consumption), as well as socioeconomically favorable indicators (e.g., high household SES, high academic achievement, perceived body image as average, and engaging in exercise). In the long-term period of COVID-19, suicidal ideation was no longer associated with following representative quarantine guidelines such as hand washing, and suicide attempts were no longer associated with internet usage time for study. Korea’s health policies within the education system have been relatively successful, but in order to prepare for the prolonged COVID-19 pandemic and/or outbreaks of new or modified infectious diseases in the future, as well as in order to manage mental health, including suicide, among Korean adolescents, it is necessary to implement a community-based management system for suicide-related mental health including for out-of-school adolescents in the future.

## Figures and Tables

**Table 1 ijerph-19-11491-t001:** General and health-related characteristics of the subjects (*N* = 167,099), *n* (%).

Variables		Total	2019 (*n* = 57,303)	2020 (*n* = 54,948)	2021 (*n* = 54,848)
Age (mean ± SD)		15.16 ± 0.01	15.08 ± 0.02	15.19 ± 0.02	15.23 ± 0.02
School type	Middle school	88,360 (49.5)	29,384 (47.9)	28,961 (49.6)	30,015 (51.0)
	High school	78,739 (50.5)	27,919 (52.1)	25,987 (50.4)	24,833 (49.0)
Gender	Male	82,679 (51.9)	28,084 (51.8)	27,199 (52.0)	27,396 (51.9)
	Female	76,510 (48.1)	25,964 (48.2)	25,295 (48.0)	5251 (48.1)
Residence with family	Yes	151,755 (96.1)	51,288 (95.6)	50,060 (96.3)	50,407 (96.4)
	No	7434 (3.9)	2760 (4.4)	23,434 (3.7)	2240 (3.6)
Household SES	Low	19,375 (11.8)	6800 (12.3)	3 (12.3)	5822 (10.7)
	Middle	77,333 (48.2)	26,049 (48.0)	25,243 (47.6)	26,041 (49.1)
	High	62,481 (40.0)	21,199 (39.6)	20,498 (40.0)	20,784 (40.3)
Subjective health status	Healthy	109,574 (68.5)	38,226 (70.4)	36,961 (70.0)	34,387 (65.2)
	Average	37,305 (23.6)	11,985 (22.4)	11,708 (22.5)	13,612 (25.9)
	Unhealthy	12,310 (7.9)	3837 (7.2)	3825 (7.5)	4648 (9.0)
Perceived body image	Thin	40,302 (25.5)	13,896 (25.8)	12,904 (24.7)	13,502 (25.8)
	Average	58,291 (36.6)	19,873 (36.6)	19,342 (36.9)	19,076 (36.3)
	Fat	60,596 (37.9)	20,279 (37.6)	20,248 (38.4)	20,069 (37.8)
Breakfast skipping	Yes	32,963 (20.6)	10,724 (19.8)	10,883 (20.5)	11,356 (21.4)
	No	126,226 (79.4)	43,324 (80.2)	41,611 (79.5)	41,291 (79.9)
Physical activity	Yes	24,035 (14.4)	8249 (14.6)	7761 (14.0)	8025 (14.6)
	No	135,154 (85.6)	45,799 (85.4)	44,733 (86.0)	44,622 (85.4)
Internet use time for study	(min)	448.74 ± 1.51	468.66 ± 2.72	417.30 ± 2.71	460.11 ± 2.34
Internet use time for leisure	(min)	201.53 ± 0.64	166.64 ± 0.83	229.37 ± 1.21	208.76 ± 1.00
Personal hygiene	Yes	59,942 (37.3)	24,599 (45.3)	16,843 (31.9)	18,500 (34.7)
	No	99,247 (62.7)	29,449 (54.7)	35,651 (68.1)	34,147 (65.3)
Current smoking	Yes	8622 (5.5)	3590 (6.9)	2476 (4.7)	2556 (5.0)
	No	150,567 (94.5)	50,458 (93.1)	50,018 (95.3)	50,091 (95.0)
Current drinking consumption	Yes	18,661 (11.8)	7694 (14.6)	5515 (10.5)	5452 (10.5)
	no	140,528 (88.2)	46,354 (85.4)	46,979 (89.5)	47,195 (89.5)

Unweighted frequency (weighted %), estimated mean ± standardized.

**Table 2 ijerph-19-11491-t002:** Suicide-related characteristics from 2019 to 2021. *n* (%).

Variables	Total	2019	2020	2021
Depression	56,794 (26.5)	14,949 (27.9)	13,100 (25.0) ***	13,933 (26.4) ***
Suicidal ideation	26,207 (12.2)	6871 (12.7)	5619 (10.7) ***	6535 (12.4)
Suicide planning	8116 (3.7)	2000 (3.6)	1794 (3.4)	2010 (3.8)
Suicide attempt	5208 (2.4)	1474 (2.6)	1007 (1.9) ***	1125 (2.1) ***

vs. 2019. *** *p* < 0.001, ** *p* < 0.01, * *p* < 0.05.

**Table 3 ijerph-19-11491-t003:** General and health-related characteristics according to the suicide-related characteristics (*N* = 159,189).

Variables		Depression		Suicidal Ideation		Suicide Planning		Suicide Attempt	
		*n* (%) or M ± SE	χ^2^ or t (*p*)	*n* (%) or M ± SE	χ^2^ or t (*p*)	*n* (%) or M ± SE	χ^2^ or t (*p*)	*n* (%) or M ± SE	χ^2^ or t (*p*)
Age		15.20 ± 0.01	12.114 ***	15.15± 0.02	−1.751	15.09 ± 0.02	6.138 ***	15.05 ± 0.02	6.545 ***
School type	Middle school	21,112 (25.0)	167.779 ***	10,380 (12.3)	22.549 ***	3384 (4.0)	71.180 ***	2164 (2.5)	66.264 ***
	High school	20,870 (27.8)		8645 (11.6)		2420 (3.2)		1442 (1.9)	
Gender	Male	17,407 (21.3)	2321.949 ***	7077 (8.7)	1678.741 ***	2267 (2.8)	335.735 ***	1181 (1.4)	478.973 ***
	Female	24,575 (31.9)		11,948 (15.4)		3537 (4.5)		2425 (3.0)	
Academic achievement	Low	16,068 (31.7)	1134.870 ***	7518 (14.7)	542.260 ***	2403 (4.7)	248.651 ***	1638 (3.1)	317.297 ***
	Middle	12,038 (25.0)		5184 (10.7)		1535 (3.2)		935 (1.9)	
	High	13,876 (23.0)		6323 (10.6)		1866 (3.1)		1033 (1.6)	
Residence with family	Yes	39,715 (26.3)	52.684 ***	17,915 (11.8)	79.998 ***	5402 (3.5)	78.463 ***	3341 (2.1)	64.275 ***
	No	2267 (30.4)		1110 (15.6)		402 (5.7)		265 (3.7)	
Household SES	Low	7092 (36.5)	1109.454 ***	3937 (20.2)	1378.292 ***	1340 (6.9)	647.865 ***	924 (4.6)	550.663 ***
	Middle	19,484 (25.4)		8577 (11.2)		2448 (3.1)		1483 (1.9)	
	High	15,406 (24.7)		6511 (10.5)		2016 (3.2)		1199 (1.9)	
Subjective health status	Healthy	23,836 (21.8)	4630.806 ***	9253 (8.5)	5543.276 ***	2636 (2.4)	2321.515 ***	1583 (1.4)	1428.590 ***
	Average	12,421 (33.1)		6143 (16.3)		1834 (4.8)		1215 (3.1)	
	Unhealthy	5725 (46.4)		3629 (29.4)		1334 (10.5)		808 (6.3)	
Perceived body image	Thin	10,423 (25.9)	163.435 ***	4539 (11.4)	383.956 ***	1327 (3.3)	120.453 ***	799 (1.9)	77.723 ***
	Average	14,527 (24.9)		6043 (10.3)		1854 (3.2)		1175 (1.9)	
	Fat	17,032 (28.2)		8443 (13.9)		2623 (4.3)		1632 (2.6)	
Breakfast skipping	Yes	9751 (29.6)	211.343 ***	4574 (13.8)	132.457 ***	1485 (4.5)	94.112 ***	977 (2.9)	94.572 ***
	No	32,231 (25.6)		14,451 (11.5)		4319 (3.4)		2629 (2.0)	
Physical activity	Yes	6452 (26.9)	3.357	2636 (11.2)	12.595 ***	914 (3.8)	2.578	596 (2.5)	13.104 **
	No	35,530 (26.3)		16,389 (12.1)		4890 (3.6)		3010 (2.1)	
Internet use time for study (min)		451.65 ± 1.55	7.392 ***	455.69 ± 1.71	8.240 ***	451.21 ± 2.39	1.346	444.76 ± 2.69	1.743
Internet use time for leisure (min)		208.72 ± 0.99	9.452 ***	212.01 ± 1.41	8.254 ***	218.33 ± 2.49	6.875 ***	219.79 ± 3.27	5.676 ***
Obesity (BMI)	Underweight	3140 (26.0)	23.085 ***	1423 (11.7)	50.173 ***	436 (3.6)	23.566 ***	259 (2.1)	2.395
	Normal	29,673 (26.8)		13,005 (11.7)		3909 (3.5)		2489 (2.2)	
	Overweight	4059 (25.4)		1937 (12.1)		630 (3.9)		378 (2.3)	
	Obese	5110 (25.6)		2660 (13.5)		829 (4.1)		480 (2.3)	
Personal hygiene	Yes	25,596 (25.9)	43.109 ***	11,212 (11.3)	109.896 ***	3540 (3.5)	6.482 *	2130 (2.1)	15.406 ***
	No	16,386 (27.4)		7813 (13.1)		2264 (3.8)		1476 (2.4)	
Current smoking	Yes	3767 (43.0)	1311.048 ***	1950 (22.0)	895.837 ***	743 (8.3)	576.287 ***	642 (7.0)	1004.595 ***
	No	38,215 (25.5)		17,075 (11.4)		5061 (3.3)		2964 (1.9)	
Current drinking consumption	Yes	7541 (40.3)	2127.632 ***	3687 (19.7)	1206.280 ***	1333 (7.0)	699.159 ***	1023 (5.3)	938.125 ***
	no	34,441 (24.5)		15,338 (10.9)		4471 (3.2)		2583 (1.8)	

*** *p* < 0.001, ** *p* < 0.01, * *p* < 0.05.

**Table 4 ijerph-19-11491-t004:** Changes between 2019 and 2020 in the relationships of the general and health related characteristics with the suicide-related characteristics.

	Depression			Suicidal Ideation			Suicide Planning			Suicide Attempt		
	OR	95% CI	*p*	OR	95% CI	*p*	OR	95% CI	*p*	OR	95% CI	*p*
**Year 2020 vs. 2019**	0.88	0.85~0.91	<0.001	0.86	0.81~0.90	<0.001	0.97	0.89~1.05	0.454	0.74	0.67~0.82	<0.001
Age	1.04	1.02~1.06	<0.001	0.98	0.95~1.00	0.066	0.95	0.91~0.99	0.007	0.91	0.86~0.95	<0.001
School type High vs. middle	0.86	0.81~0.92	<0.001	0.74	0.68~0.81	<0.001	0.65	0.56~0.75	<0.001	0.61	0.51~0.74	<0.001
Gender Female vs. male	1.86	1.79~1.93	<0.001	1.92	1.82~2.02	<0.001	1.65	1.27~1.80	<0.001	2.44	2.20~2.71	<0.001
Academic achievement Middle vs. low	1.07	1.03~1.12	<0.001	0.99	0.94~1.05	0.725	1.03	0.93~1.13	0.621	1.15	1.02~1.29	0.024
High vs. low	1.32	1.26~1.37	<0.001	1.22	1.15~1.28	<0.001	1.30	1.18~1.43	<0.001	1.49	1.32~1.67	<0.001
Living with family Yes vs. no	0.87	0.80~0.94	<0.001	0.75	0.68~0.83	<0.001	0.60	0.51~0.70	<0.001	0.52	0.48~0.70	<0.001
Household SES Middle vs. low	0.91	0.87~0.94	<0.001	0.96	0.91~1.01	0.093	0.88	0.81~0.96	0.004	0.86	0.77~0.95	0.005
High vs. low	1.34	1.27~1.41	<0.001	1.66	1.56~1.77	<0.001	1.61	1.45~1.80	<0.001	1.68	1.47~1.91	<0.001
Subjective health status Average vs. healthy	1.66	1.60~1.72	<0.001	1.90	1.81~1.99	<0.001	1.87	1.72~2.04	<0.001	1.95	1.75~2.17	<0.001
Unhealthy vs. healthy	2.82	2.67~2.98	<0.001	3.93	3.68~4.19	<0.001	4.12	3.73~4.54	<0.001	4.13	3.66~4.66	<0.001
Perceived body image Average vs. thin	0.90	0.87~0.95	<0.001	0.92	0.86~0.97	0.006	1.01	0.91~1.12	0.878	0.94	0.83~1.07	0.379
Fat vs. thin	1.04	0.99~1.09	0.105	1.13	1.06~1.22	0.001	1.23	1.09~1.37	0.001	1.25	1.08~1.44	<0.001
Breakfast skipping Yes vs. no	1.12	1.08~1.16	<0.001	1.10	1.05~1.16	<0.001	1.16	1.07~1.26	<0.001	1.22	1.10~1.34	<0.001
Physical activity Yes vs. no	1.29	1.24~1.35	<0.001	1.19	1.12~1.27	<0.001	1.24	1.11~1.38	<0.001	1.49	1.32~1.69	<.001
Internet use time for study (min)	1.01	1.01~1.02	<0.001	1.02	1.02~1.03	<0.001	1.02	1.01~1.03	0.003	1.00	0.99~1.01	0.879
Internet use time for leisure (min)	1.01	1.01~1.02	<0.001	1.02	1.01~1.02	<0.001	1.02	1.00~1.03	0.012	1.02	1.00~1.04	0.040
BMI Normal vs. underweight	1.14	1.06~1.22	<0.001	1.07	0.98~1.18	0.132	1.04	0.89~1.22	0.621	1.10	0.90~1.34	0.343
Overweight vs. underweight	1.01	0.92~1.11	0.080	1.02	0.90~1.15	0.738	1.03	0.84~1.26	0.778	1.03	0.80~1.32	0.824
Obese vs. underweight	0.94	0.87~1.03	0.190	1.07	0.95~1.20	0.283	1.01	0.82~1.23	0.958	1.01	0.79~1.30	0.942
Personal hygiene Yes vs. no	1.02	0.98~1.05	0.316	1.09	1.04~1.14	<0.001	0.99	0.92~1.07	0.829	0.99	0.90~1.08	0.763
Smoking Yes vs. no	1.66	1.55~1.79	<0.001	1.79	1.64~1.96	<0.001	1.99	1.74~2.29	<0.001	2.87	2.46~3.36	<0.001
Drinking consumption Yes vs. no	1.78	1.69~1.87	<0.001	1.81	1.69~1.93	<0.001	1.89	1.70~2.10	<0.001	2.31	2.04~2.63	<0.001

BMI: Body mass index.

**Table 5 ijerph-19-11491-t005:** Changes between 2019 and 2021 in the relationships of the general and health-related characteristics with the suicide-related characteristics.

	Depression			Suicidal Ideation			Suicide Planning			Suicide Attempt		
	OR	95% CI	*p*	OR	95% CI	*p*	OR	95% CI	*p*	OR	95% CI	*p*
**Year 2021 vs. 2019**	0.91	0.88~0.95	<0.001	0.96	0.92~1.01	0.102	1.04	0.96~1.12	0.336	0.80	0.72~0.87	<0.001
Age	1.03	1.01~1.05	0.001	0.96	0.94~0.98	<0.001	0.91	0.88~0.95	<0.001	0.87	0.83~0.92	<0.001
School type High vs. middle	0.83	0.78~0.88	<0.001	0.71	0.65~0.77	<0.001	0.68	0.59~0.78	<0.001	0.64	0.54~0.76	<0.001
Gender Female vs. male	1.76	1.70~1.82	<0.001	1.94	1.85~2.03	<0.001	1.65	1.52~1.79	<0.001	2.30	2.08~2.54	<0.001
Academic achievement Middle vs. low	1.09	1.05~1.13	<0.001	1.01	0.96~1.07	0.661	0.99	0.90~1.09	0.870	1.14	1.02~1.28	0.024
High vs. low	1.34	1.29~1.40	<0.001	1.18	1.12~1.25	<0.001	1.15	1.05~1.26	0.003	1.40	1.22~1.58	<0.001
Living with family Yes vs. no	0.83	0.77~0.89	<0.001	0.68	0.62~0.76	<0.001	0.54	0.47~0.64	<0.001	0.52	0.44~0.63	<0.001
Household SES Middle vs. low	0.88	0.85~0.92	<0.001	0.96	0.91~1.01	0.097	0.89	0.81~0.96	0.004	0.86	0.78~0.96	0.006
High vs. low	1.31	1.24~1.38	<0.001	1.68	1.57~1.79	<0.001	1.72	1.55~1.91	<0.001	1.79	1.58~2.02	<0.001
Subjective health status Average vs. unhealthy	1.70	1.64~1.76	<0.001	1.94	1.86~2.03	<0.001	1.94	1.78~2.10	<0.001	1.96	1.77~2.17	<0.001
Unhealthy vs. healthy	2.83	2.69~2.98	<0.001	4.07	3.82~4.33	<0.001	4.35	3.96~4.79	<0.001	3.94	3.50~4.44	<0.001
Perceived body image Average vs. thin	0.91	0.87~0.95	<0.001	0.89	0.84~0.95	<0.001	1.04	0.93~1.16	0.514	1.06	0.92~1.21	0.435
Fat vs. thin	1.05	1.00~1.11	0.056	1.11	1.03~1.19	0.004	1.15	1.03~1.29	0.018	1.21	1.04~1.41	0.012
Breakfast skipping Yes vs. no	1.11	1.07~1.16	<0.001	1.12	1.07~1.18	<0.001	1.20	1.11~1.30	<0.001	1.28	1.16~1.40	<0.001
Physical activity Yes vs. no	1.31	1.25~1.37	<0.001	1.24	1.17~1.32	<0.001	1.36	1.23~1.50	<0.001	1.48	1.31~1.68	<0.001
Internet use time for study (min)	1.01	1.00~1.01	<0.001	1.02	1.01~1.02	<0.001	1.01	1.00~1.02	<0.001	1.00	0.99~1.02	0.621
Internet use time for leisure (min)	1.02	1.01~1.02	<0.001	1.02	1.01~1.03	<0.001	1.03	1.01~1.04	<0.001	1.01	0.99~1.03	0.237
BMI Normal vs. underweight	1.12	1.05~1.20	0.001	1.08	0.98~1.18	0.105	0.91	0.78~1.06	0.204	0.98	0.80~1.19	0.821
Overweight vs. underweight	0.97	0.89~1.06	0.530	1.06	0.94~1.20	0.333	0.96	0.78~1.17	0.651	0.97	0.75~1.25	0.893
Obese vs. underweight	0.91	0.84~0.99	0.027	1.06	0.94~1.19	0.329	0.92	0.75~1.12	0.393	0.92	0.71~1.19	0.533
Personal hygiene Yes vs. no	0.96	0.93~0.99	0.020	1.04	0.99~1.08	0.114	0.94	0.87~1.02	0.143	0.97	0.89~1.06	0.483
Smoking Yes vs. no	1.63	1.52~1.75	<0.001	1.91	1.75~2.09	<0.001	1.94	1.70~2.22	<0.001	2.65	2.26~3.10	<0.001
Drinking consumption Yes vs. no	1.80	1.72~1.89	<0.001	1.74	1.64~1.86	<0.001	2.11	1.90~2.34	<0.001	2.47	2.18~2.81	<0.001

BMI: Body mass index.

## Data Availability

The datasets used and/or analyzed during the current study are available from the corresponding author on reasonable request.

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
