# Peer review of "Short- and Long-Term Impacts of the COVID-19 Pandemic on Suicide-Related Mental Health in Korean Adolescents"

_ijerph, 2022, doi:10.3390/ijerph191811491_

Round 1

Reviewer 1 Report

This paper investigates the short-term and long-term impacts of the COVID-19 pandemic on suicide-related characteristics in South Korean adolescents by comparing the data from 2020 and 2021 with the data from 2019. The study was based on a Web-based survey targeting school students. They found significant reduction depression and suicidal attempts during the pandemic.

This is an attractive article given that COVID-19 crisis poses the great threat to mental health. However, I feel that the authors need to justify the results in detail. I have the following concerns.

Add a discussion about why the suicide-related characteristics in South Korean adolescents have been improved during the pandemic.

I am not sure if the study appropriately represented the South Korean adolescent population. Despite the changes during the pandemic, the sample sizes were roughly the same in the three years. Provide the details about how the participants of the KYRBS survey were selected.

The results contradict to several articles from South Korean newspapers and research papers. For example, Korea Herald reported that “suicides among young people increased during the first year of the COVID-19 crisis in 2020, despite a drop overall in South Korea.   In 2020, 957 children and adolescents ages 9-24 took their own lives -- almost 2 per day -- indicating an increase for the fourth year in a row. For the three preceding years from 2017 to 2019, the number of suicides in this age group was 722, 827 and 876, respectively.” See the following link.

(https://www.koreaherald.com/view.php?ud=20220614000856)

Among several others, the following study shows similar results by analyzing the suicide rate by age groups.

https://www.statista.com/statistics/789375/south-korea-suicide-death-rate-by-age-group/

I believe the number of suicide attempts is highly correlated with the number of suicide deaths.

Reviewer 2 Report

The subject discussed by the authors is very important due to the impact of the epidemic on the mental health of adolescents.

This difficult period for everyone had an impact on the described research group, an other adolescents globally, which could be observed in the increased number of psychiatric and psychological interventions among adolescents.

I propose to make a few minor changes.

Tables should only be included in the Results section, not in the Discussion.

The authors have collected extensive literature on similar topics, but have not used it fully in the area of discussing the results achieved. I propose to expand this part of the article based on the cited publications.

I also suggest describing the proposed therapeutic interactions in more detail in terms of the discussed difficulties in the mental functioning of this age group. Mainly in the context of young people's isolation from their peers.

Round 2

Reviewer 1 Report

The issues raised on the previous submission have been addressed appropriately.